# Cancer Research Trend Analysis Based on Fusion Feature Representation

**DOI:** 10.3390/e23030338

**Published:** 2021-03-12

**Authors:** Jingqiao Wu, Xiaoyue Feng, Renchu Guan, Yanchun Liang

**Affiliations:** 1Zhuhai Sub Laboratory of Key Laboratory of Symbolic Computation and Knowledge Engineering of the Ministry of Education, Zhuhai College of Jilin University, Zhuhai 519041, China; wujingqiao17@gmail.com (J.W.); guanrenchu@jlu.edu.cn (R.G.); 2Key Laboratory of Symbolic Computation and Knowledge Engineering of the Ministry of Education, College of Computer Science and Technology, Jilin University, Changchun 130012, China; fengxy@jlu.edu.cn

**Keywords:** feature representation, feature fusion, trend analysis, text mining

## Abstract

Machine learning models can automatically discover biomedical research trends and promote the dissemination of information and knowledge. Text feature representation is a critical and challenging task in natural language processing. Most methods of text feature representation are based on word representation. A good representation can capture semantic and structural information. In this paper, two fusion algorithms are proposed, namely, the Tr-W2v and Ti-W2v algorithms. They are based on the classical text feature representation model and consider the importance of words. The results show that the effectiveness of the two fusion text representation models is better than the classical text representation model, and the results based on the Tr-W2v algorithm are the best. Furthermore, based on the Tr-W2v algorithm, trend analyses of cancer research are conducted, including correlation analysis, keyword trend analysis, and improved keyword trend analysis. The discovery of the research trends and the evolution of hotspots for cancers can help doctors and biological researchers collect information and provide guidance for further research.

## 1. Introduction

Since the completion of the Human Genome Project and with the rapid development of high-throughput biotechnology, the amount of data in the fields of biology, medicine, genetics, and chemistry has exponentially grown. As of January 2021, the number of entries in PubMed (Biomedical Literature Retrieval System) has exceeded 30 million [1]. However, given the large-scale, rapid growth and massive amounts of data in various formats, people can do little with the data. It is a major challenge for clinicians or biological researchers to obtain cutting-edge information about research from tens of thousands of publications. Traditional methods, the knowledge of which was manually acquired from literature and images, can no longer meet researchers’ needs for understanding the current hotspots and trends of biomedical research [2,3]. It has become urgent to use intelligent algorithms to quickly and effectively acquire and discover biomedical knowledge. 

Cancer research has attracted much attention in the medical field. Among all cancers, lung cancer poses the greatest threat to human health; it is characterized by its rapid spread and high probability of death. In recent years, according to statistical data around the world, the possibility of people suffering from lung cancer has greatly increased. Besides, gastric cancer, colorectal cancer, breast cancer, and liver cancer are also high-risk cancers that have been studied in the medical field in recent years. Traditional trend analysis can only be completed after reading and sorting out many documents published in the field in recent years by experts. This approach may hinder the dissemination of information and knowledge and cause omissions in the retrieval of papers by experts, which may affect the results of the extraction of research hotspots or trend analysis. The usage of machine learning models to automatically discover biomedical research trends can make up for this deficiency [4,5].

Text feature learning is an important task in the field of natural language processing, and it is the basis of many downstream applications, such as text clustering and classification [6]. Most existing text feature representation learning is based on words, that is, word vector representation. It obtains word representation by mapping words from a one-dimensional space to a continuous vector space. The word representation methods include neural networks, word co-occurrence methods, methods that rely on probability, and interpretable knowledge base methods. A good low-dimensional mapping representation often improves the performance of downstream tasks [7]. Feature fusion is the integration of multiple different feature information to obtain more prominent feature information [8,9,10,11]. Multimodal features from text, audio and vision can be fused with fusion technique [12]. There are two types of fusion technique, early fusion, and late fusion. Early fusion concatenates the features together at first and late fusion combines results [13]. We adopt early fusion for text clustering. 

Based on text representation methods, we propose a multi-view feature fusion strategy. The hotspots and trend analysis were conducted on 260,000 cancer studies using the proposed method. Our contribution mainly includes the following points. (1) The fusion of the improved vector representation model Ti-W2v algorithm and Tr-W2v algorithm were proposed. (2) A correlation analysis algorithm based on similarity is proposed to analyze the relationship among five cancer types. (3) A keyword trend analysis model and its improved model are proposed. Taking lung cancer as an example, the keyword analysis model analyzes the overall research hotspots. (4) Taking lung cancer as an example, the trend of lung cancer research is further analyzed from three perspectives, including gene proteins, therapeutic drugs and methods. The results can help guide the literature summary and further work of relevant researchers.

The remainder of this paper is organized as follows: Section 2 lists the materials and methods. Section 3 describes the experimental details, presents the experimental results, and gives the error analysis. Section 4 discusses the results. Section 5 concludes our work.

## 2. Materials and Methods

### 2.1. Background

Traditional text representation models commonly include models based on word frequency, TF-IDF, TextRank, and word embedding. The text feature representation model based on word frequency is the simplest. It calculates the number of occurrences of each word in the text and obtains the text vector with the word frequency of each word [14]. The expression based on the word frequency algorithm is shown in Equation (1):(1)$$wordcount\left(i,j\right)=n_{i,j}$$
where *n_i,j_* is the occurrence number of word *t_i_* in document *d_j_*.

The text feature representation model based on TF-IDF considers the frequency of occurrence of each word in the training texts and the number of other training text containing the word, that is, the frequency of the reverse text [15,16]. The expression of the TF-IDF algorithm is shown as Equation (2):(2)$$tfidf\left(i,j\right)=tf\left(i,j\right)*idf\left(i\right)\;=\frac{n_{i,j}}{\sum_{k}n_{k,j}}*log\frac{\left|D\right|}{1+\left|\left\{j:t_{i}\epsilon d_{j}\right\}\right|}$$
where |*D*| represents the total number of files in the corpus. $1+\left|\left\{j:t_{i}\in d_{j}\right\}\right|$ represents numbers of documents containing the term $t_{i}$, we add 1 here to prevent the denominator from being 0. The TF-IDF text representation model is an algorithm based on word frequency, which pays more attention to the number of times the words appeared in the document and does not consider the relative position between them.

The TextRank-based text feature representation model is a graph-based sorting algorithm for text [17]. Its core idea is that a word is more important if it appears after many words. Besides, if a word is followed by another word with a high TextRank value, the TextRank value of this word is accordingly higher. The TextRank model is an algorithm based on graphs. Let G = (*V*,*E*) be a directed graph with the set of vertices *V* and set of edges *E*, where *E* is a subset of *V*V*. For a given vertex *V_i_*, let *In*(*V_i_*) be the set of vertices that point to it (predecessors), and let *Out*(*V_i_*) be the set of vertices that vertex *V_i_* points to (successors). The score of a vertex *V_i_* is defined as followed Equation (3):(3)$$WS\left(V_{i}\right)=\left(1-d\right)+d*\underset{V_{j}\epsilon In\left(V_{i}\right)}{\sum }\frac{w_{ji}}{\sum_{V_{k\epsilon Out\left(V_{j}\right)}}w_{jk}}WS\left(V_{j}\right)$$
where *d* is a damping factor that can be set between 0 and 1; *w_ji_* is the weight between *V_j_* and *V_i_*. The TextRank model focuses more on the degree of co-occurrence between words in a fixed-length window. This considers the relative position of words to a certain extent, so when the number of documents is small, the TextRank algorithm can express text information more accurately, while the TF-IDF algorithm cannot do this.

The text feature representation model based on word embedding maps words to another space through a certain mapping rule and generates expressions in a new space [18]. The word embedding text representation model is an algorithm based on a neural network. The hidden attributes between words in the text, such as the similarity and part of speech between words, are emphasized. As the characteristics of neural networks, the word embedding text representation model is difficult to be explained, but its final effect is better than TF-IDF and the TextRank algorithm. The obtained word vectors can measure the semantic and other relevant features between words. Therefore, word embedding methods to represent text features has been a hotspot in recent years. The most commonly used word embedding tool is Word2Vec [19,20,21], which contains two training modes: the CBOW training mode and the Skip-gram training mode.

### 2.2. Method

As shown in Figure 1, our proposed framework consists of two modules, a feature fusion module, and a research trend analysis module. The feature fusion module contains two fusion strategies, and the research trend analysis includes three trend analysis methods.

#### 2.2.1. Feature Fusion Representation Model

The word-based text representation method needs to obtain the representation of each word first. Then, word vectors can be used to obtain a text representation. The classic method is used to add all word vectors and the average of all vectors as the text vector. In this method, all words in the text are considered equally important. This is obviously far-fetched because the importance of words in the text is different. The representation algorithms of TF-IDF and TextRank represent a text by calculating the weight of words in the text, but the analysis point and calculation method of the two are quite different. The Word2Vec algorithm can determine the semantic information of words and does not consider the importance of words. To retain the advantages of the above methods, we propose a multi-view fusion strategy, which combines Word2Vec with TF-IDF and TextRank. In this fusion strategy, Word2Vec is chosen as the representation method of words. The weights of words in text are given by TF-IDF and TextRank. We named the fusion method Ti-W2v and Tr-W2v, and the details are given in the following sections.

Ti-W2v is an improved algorithm that combined TF-IDF and Word2Vec. TF-IDF is adopted to calculate the weight coefficient of each word in the text, and the embedding vector of the text can be generated with the product of the weights and embedding vectors of Word2Vec for all words. The advantage is that different words in the text can be given different degrees of importance, closer to the actual situation than average embedding. For a corpus, D is the corpus, and D = {*D*_1_, *D*_2_, …, *D_k_*}, *D_i_* is the *i*th document. *V_i_* is the representation of *D_i_*. *w_ij_* is the *j*th word in *D_i_*, *v_ij_* is the vector of *w_ij_* obtained by word2vec. *TI_ij_* is the weight of *w_ij_*, obtained with TF-IDF as Equation (4): (4)$$TI_{ij}=\frac{n_{i,j}}{\sum_{k}n_{k,j}}*log\frac{\left|D\right|}{1+\left|\left\{j:t_{i}\in d_{j}\right\}\right|}$$
*V_i_* is defined using Equation (5):*V_i_* = *TI_ij_* × *v_ij_*(5)

The TF-IDF algorithm is based on word frequency. It measures the importance of words based on text word frequency and global reverse text frequency. It is suitable for cases in which the number of documents is relatively large. While, in TextRank, the importance of words is decided by their relative position. It does not depend on other documents and considers the co-occurrence of each word. Based on the fusion strategy, we propose the Tr-W2v algorithm, which combines TextRank and Word2Vec. First, TextRank is used to calculate the weight coefficients of different words in the text, and then the Word2Vec embedding vectors of the words by weight are added to obtain the text vector. *TR_ij_* is the weight of *w_ij_*, obtained with TextRank as Equation (6): (6)$$TR_{ij}=\left(1-d\right)+d*\underset{V_{m}\epsilon In\left(V_{j}\right)}{\sum }\frac{w_{mi}}{\sum_{V_{k\epsilon Out\left(V_{m}\right)}}w_{mk}}WS\left(V_{m}\right)$$
as Ti-W2v, *V_i_* is defined with Equation (7):*V_i_* = *TI_ij_* × *v_ij_*(7)

#### 2.2.2. Cancer Research Trend Analysis Model

Based on the fusion-improved feature representation model proposed in the previous section, we propose three trend analysis models. We first propose a similarity trend analysis model based on the five high-incidence cancer datasets. A keyword trend analysis model and an improved keyword analysis model are proposed based on the lung cancer dataset. Then, lung cancer-related gene proteas, treatment methods, and drugs, and other hotspots related to lung cancer were analyzed.

##### Correlation Analysis Based on Similarity

We use the Tr-W2v algorithm to obtain the corresponding text vectors of abstracts on the five major cancer in the last five years. Then, the text vectors of various cancers are integrated into a vector for a certain year of this type of cancer in units of years (addition and average). The cosine similarities of different cancers are calculated in different years, and the correlation of different cancers are analyzed for the past five years through cosine similarity. Figure 2 shows the flowchart of the algorithm.

##### Keyword Trend Analysis Model

Taking the lung cancer dataset as an example, we use the TextRank algorithm to obtain the top 10% of keywords and corresponding weights in each document. Then, all the keywords and corresponding weights of the year are integrated into units of years. The method of integration is as follows: for the keywords that have not appeared, we add them and the corresponding weights directly to the keywords of the year. For the keywords that have appeared, we add and merge their weights as their new weights. Finally, the top 50 keywords were obtained as hotspots of the year through keyword reordering. Figure 3 shows the flowchart of the algorithm.

##### Improved Keyword Trend Analysis Model

The keyword analysis model proposed in the previous section can coarsely analyze the annual research hotspots of single types of cancer (taking lung cancer as an example). For more detailed trend analysis, we propose an improved keyword analysis on this basic model. As in the correlation analysis, we first use the Tr-W2v algorithm to obtain the text vector corresponding to lung cancer of each year. Further, the k-means clustering algorithm is adopted, and *k* categories are generated. The keyword integration operation in the previous section is utilized to obtain hotspots of different clusters. Then, the top keyword of each category is extracted and integrated into the distribution of hotspots of that year. Figure 4 shows flowchart of the algorithm.

## 3. Results

### 3.1. Datasets

For comparing the effect of representation methods, we use the second edition of the well-known public classification dataset 20 newsgroups [22]. In the analysis of cancer research trends, we retrieve PubMed articles using MeSH terms and obtain experimental datasets that include data from the past five years on the five most common cancers in China (lung cancer, breast cancer, gastric cancer, colorectal cancer, and liver cancer) [23,24]. Table 1 shows the distribution of the number of research papers for the five major cancers in the most recent five years.

### 3.2. Results

In the cancer dataset, the most papers on lung cancer and breast cancer were published in 2018. The number of papers published in 2014 is the largest for gastric cancer. For colorectal and liver cancer, the number of papers published in 2015 is the largest. The number of cancer papers has not increased over the years. It shows a stable trend, and in some years, the trend is slightly lower than in previous years; however, the total number of cancer research papers is still rising slightly.

#### 3.2.1. Comparison Results of Feature Fusion Methods

To compare the results of feature fusion methods, we conduct clustering experiments on five text representation algorithms, including TF-IDF, Word2Vec, TextRank, Ti-W2v, and Tr-W2v. First, the five algorithms are used to vectorize the text of the data set. Then, we use the classical k-means clustering algorithm to evaluate the effects of the five word-representation algorithms. We select ten categories from 20 newsgroups dataset as the experimental dataset. The number of clusters in k-means is set to 10, the initialization method defaults to k-means++, and the maximum iteration number is set to 300. Using the silhouette coefficient of clustering as a measurement [25]. The result of TF-IDF, Word2Vec, TextRank, Ti-W2v, and Tr-W2v are 0.402, 0.449, 0.433, 0.491, and 0.502, respectively. Figure 5 shows the two-dimensional clustering visualization effect of the data.

In the clustering experiment, the clustering silhouette coefficients based on TF-IDF, Word2Vec, TextRank, Ti-W2v, and Tr-W2v are 0.402, 0.449, 0.433, 0.491, and 0.502, respectively. Among them, the Tr-W2v algorithm has the best result. The effect of Word2Vec vector is 11.7% higher than that of TF-IDF. The effect of the TextRank vector is 7.7% higher than that of TF-IDF. Ti-W2v has a 9.4% improvement over Word2Vec. Tr-W2v has a 2.2% improvement over Ti-W2v. The choice of word vectors plays a vital role in text representation. It is best to use the Word2Vec method to improve the results. Additionally, the word vector fusion method also has a certain effect. The effect based on the TextRank fusion text vector is better than that of the TF-IDF fusion text vector. They are both better than Word2Vec. Although the improvement effect is not as obvious as the replacement of word vectors, there is also a certain degree of improvement. In general, Tr-W2v fusion text vectors have the best clustering effect, which also reflects that it can better represent the text information. We evaluated the effect of the Tr-W2v algorithm’s TextRank window size. When the window size is 2, 4, 5, 6, 10, the results are 0.469, 0.485, 0.502, 0.490, and 0.453, respectively. We choose 5 as the window size.

The results of the fusion feature experiment show that the fusion vector obtained by the Tr-W2v algorithm has the best result in clustering experiments, and the Ti-W2v algorithm is slightly inferior to the Tr-W2v algorithm; however, both are better than the traditional text representation model. It may be caused by the difference between the TF-IDF algorithm and the TextRank algorithm. The TF-IDF algorithm only considers word frequency information and does not consider the relationship between words. Compared with the TF-IDF algorithm, TextRank can obtain important information, such as the relative position of words within a single text, so the integration of TextRank and Word2Vec will perform better. In addition, we can see that the TF-IDF and TextRank vector clustering effects are slightly different from the other three vector clustering effects. The TF-IDF and TextRank algorithms represent vectors by word frequency and word co-occurrence position, respectively. Meanwhile, the other three algorithms are based on Word2Vec’s low-dimensional dense vectors. Therefore, the clustering shape based on TF-IDF and the TextRank algorithm are more decentralized, while the other three algorithms are more uniform and regular.

#### 3.2.2. Cancer Trend Analysis Results

Next, the experimental results of the cancer research trend analysis model based on the fusion-improved feature representation model are listed below.

##### Correlation Analysis Results Based on Similarity

A correlation analysis is conducted based on similarity to determine the relationships among the five cancer types. Figure 6 shows the results for the most recent five years.

From Figure 6, we can conclude that colorectal cancer is most closely related to the other four cancers. The following reasons indicate that smoking may cause lung cancer; long-term smokers are more likely to die from colorectal cancer than nonsmokers [26]. There are many repeated research directions for the treatment of breast cancer and colorectal cancer [27,28]. The stomach and colorectal are organs of the digestive tract system, and many studies are conducted simultaneously [29,30]. The above studies can confirm the close relationship between colorectal cancer and four other cancers from the side. Lung cancer, breast cancer, gastric cancer, and colorectal cancer have the lowest similarity with liver cancer. In addition, lung cancer has the highest similarity to colorectal cancer among all relations, and breast cancer has the lowest similarity to liver cancer. This also shows that among the top five high-incidence cancers, lung cancer is most closely linked to colorectal cancer, while breast cancer is relatively less linked to liver cancer.

##### Results of the Keyword Trend Analysis Model

Taking lung cancer as an example, Figure 7 shows the visualization results of the annual hotspots word cloud of lung cancer obtained by the keyword trend analysis model and the improved keyword trend analysis model.

It can be found that the hotspots in the past five years have focused on the *patient*, *cancer*, *cell*, *lung*, *study*, *tumor*, *CI*, *nsclc*, and so on. According to the principles of the TextRank algorithm and the characteristics of the lung cancer research literature, the above results are normal because the central idea of the TextRank algorithm is that the more times a word and other important words co-occurrence within a certain length, the more important the word is. The above vocabulary in the literature of lung cancer research uses TextRank’s weight ranking mechanism, and the above vocabulary may exist in the important vocabulary and hotspot area of the study in the literature on lung cancer. Public hotspots are basically the same each year using the keyword trend analysis model. The results are too rough to study the trend of lung cancer in the past five years. The improved keyword trend analysis model is optimized with more details. The results are significantly different from the keyword analysis methods before improvement (see the right-hand side of Figure 7). Based on improved methods, hotspots in different categories are combined to generate hotspots of one year. Then, the differential hotspots between years are chosen to represent the public hotspot of each year. Therefore, the hotspots of each year are clearly distinguished, which makes it easier and clearer to analyze the research trends of lung cancer in the recent five years.

##### Results of Analysis on Research Trend

Based on improved keyword trend analysis results, to present the research trend on lung cancer, research trends in different areas are listed in Figure 8, Figure 9 and Figure 10. Figure 8 shows the research trends of related gene proteins and invertase factors in lung cancer research in the past five years. Figure 9 shows the hot research trends of lung cancer-related treatment drugs and methods in the past five years, and Figure 10 shows the other hot topics of lung cancer in the past five years’ research trends.

The number of genes and proteins in the human body is very large, and many studies invest in gene protein-related research on lung cancer each year. It can be seen from Figure 8 that the research hotspots for genes and proteins are various in different years. The unique hot research terms in 2014 included the ATK1 gene, YAP gene, PKM2 gene, LSCC gene, and PDCD5 gene. The unique hot research terms in 2015 included PMS separation enhancer protein, BBP gene, Bsm gene, and THOR long noncoding RNA. The unique hot research terms in 2016 included SFTPD gene, p110α protein, DDX17 gene, Globo H glycoprotein, and LHX6 gene. The unique hot research terms in 2017 included SiRNA, TGF-β transforming growth factor, luciferase, RDM1 protein, and SNHG15 long-chain noncoding RNA. The unique hot research terms in 2018 included miRNA-223 and LKB1 gene.

With the development of science and technology, different lung cancer related treatment drugs and treatment methods have emerged. It can be seen from Figure 9 that the research hotspots for different drugs and treatments vary by year. The unique research words in 2014 included metal platinum anticancer drugs CHIP, galactosylceramide, thymoquinone drugs, AR lung nodule intelligent diagnosis system treatment, NC treatment, and TP treatment. The unique research hotspot words in 2015 included enzalutamide drugs, tenofovir dipivoxil drugs, dibenzylthiocaprylic acid drugs, thalidomide drugs, PEGPH2O tumor effect drugs, EP regimen treatment, intraoperative radiotherapy, and robot-assisted thoracoscopic surgery. The unique research hotspot words in 2016 included linsitinib drugs, penicillin drugs, SKLB drugs, sitagliptin drugs, erlotinib drugs, statins, ARMS quantitative treatment, and PCR-clamp method detection. The unique research hotspot words in 2017 included cucurbitacin, human resistin, and dimethyl amiloride. The unique research hotspot words in 2018 included aspirin drugs, intraoperative radiotherapy for lung cancer, gamma knife treatment, leishmaniasis, and low-dose lung CT technology.

It can be seen from Figure 10 that the research hotspots for other factors related to lung cancer in different years are also different. The unique hot research terms in 2014 included DOX, derivative, exosomes, and mesothelioma. The unique hot research terms in 2015 included DFI and bmplc. The unique hot research terms in 2016 included TMB tumor mutation load, monocyte leukemia CMML, and Co3O4 nanoparticles. The hot research terms in 2017 included adhesion, csc, and tam. The unique hot research terms in 2018 included image, examination, software, feature, algorithm, epithelioid, pneumonia, and bilateral. Regarding the hot words in 2018, it is noteworthy that with the advancement of science and technology, computer software and artificial intelligence algorithms play an increasingly important role in lung cancer research, such as artificial intelligence for image diagnosis of the lung.

## 4. Limitation

There are some limitations in our work. We only took five cancers as examples and discussed their relevance. More cancer data should be added. Further, the word2vec is chosen as the embedding method. Advanced text representation methods such as BERT (Bidirectional Encoder Representations from Transformers) [31] and BioBERT [32] might be a better choice. For the design of the experiment, we apply three methods to analyze the trend of cancer, and more diversified test methods could be used in future work.

## 5. Conclusions

Text feature representation models play an essential role in natural language processing. Improving these models helps machines better understand relevant text information and promote downstream tasks. Considering the words’ degree of importance, we combined the TF-IDF and TextRank with word2vec. Results demonstrate the effectiveness of the fusion models. Meanwhile, the combined model is adopted to present research trend analysis of cancers. The proposed models can help researchers find research hotspots in biology, medicine, information retrieval, and natural language processing.

## Figures and Tables

**Figure 1 entropy-23-00338-f001:**
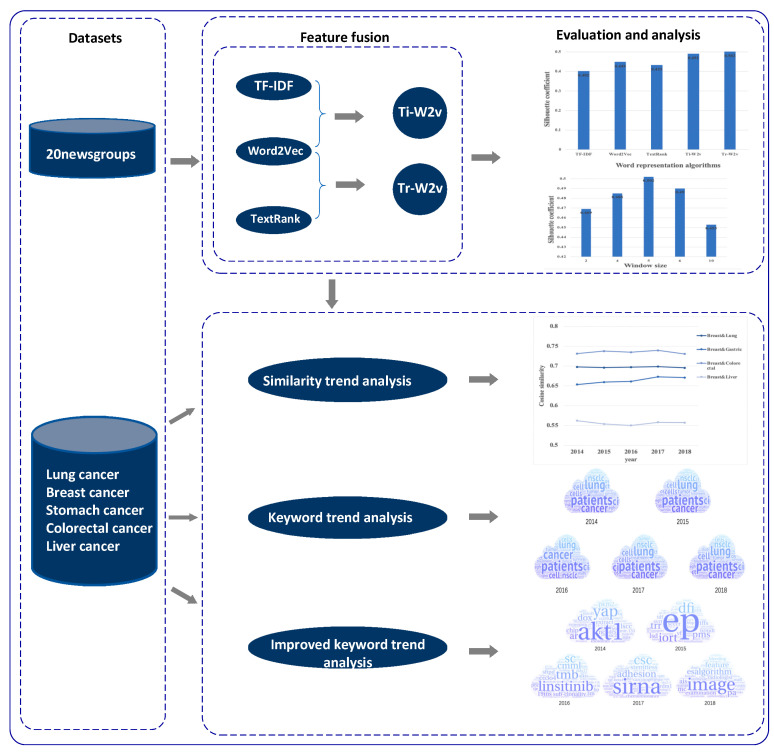
Framework of our work.

**Figure 2 entropy-23-00338-f002:**
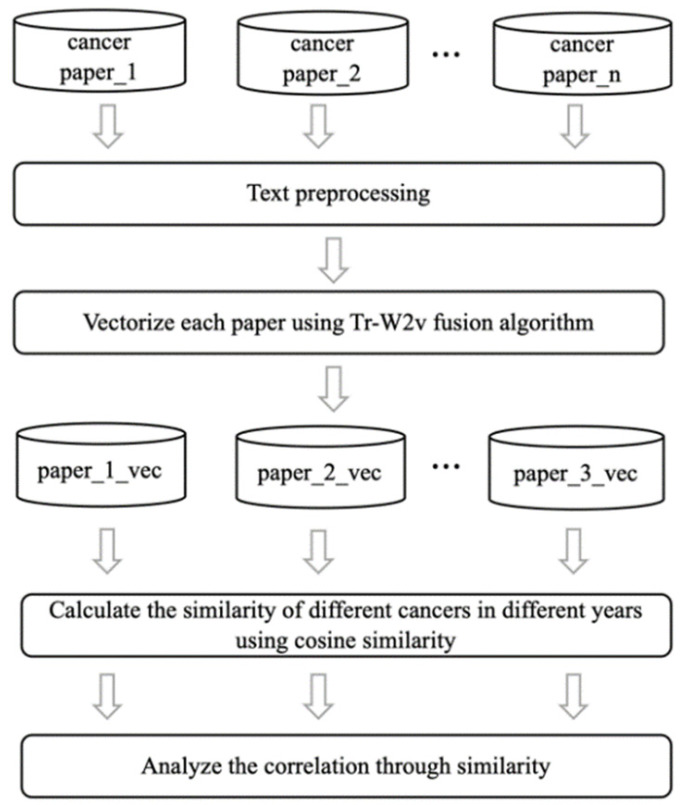
Flowchart of similarity trend analysis.

**Figure 3 entropy-23-00338-f003:**
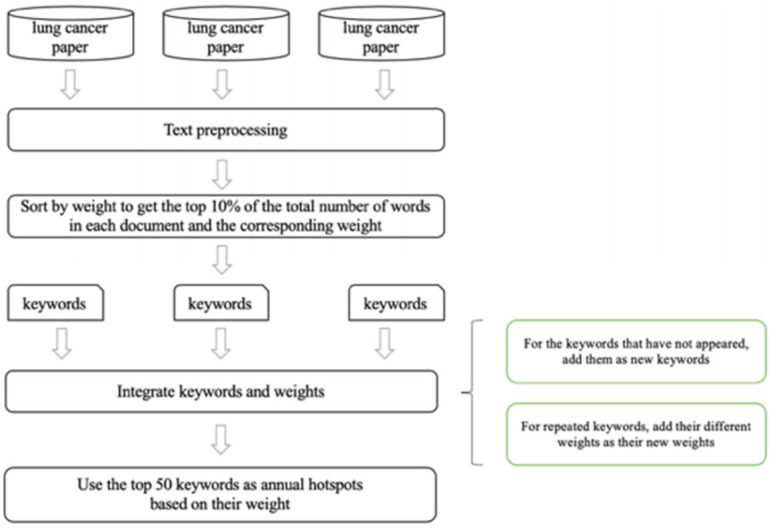
Flowchart of keyword trend analysis.

**Figure 4 entropy-23-00338-f004:**
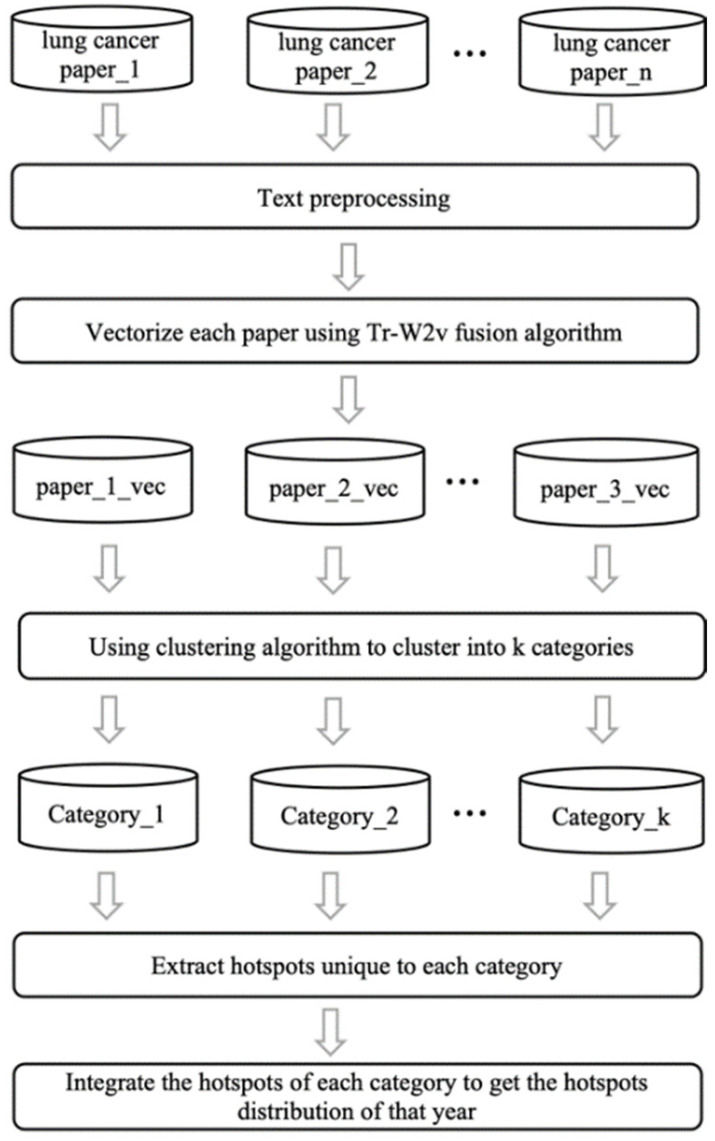
Flowchart of improved keyword trend analysis.

**Figure 5 entropy-23-00338-f005:**
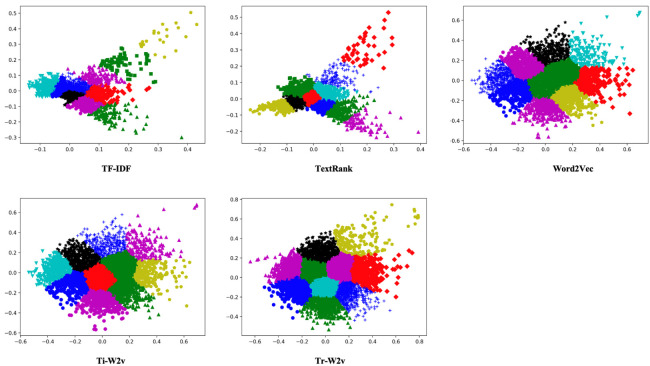
Two-dimensional clustering visualization results based on five word-representation algorithms.

**Figure 6 entropy-23-00338-f006:**
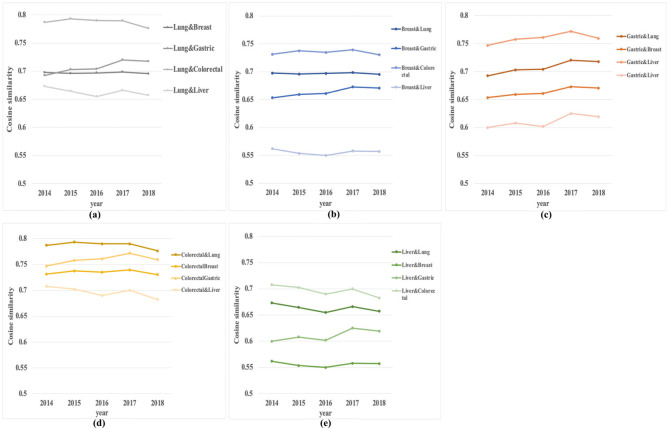
Correlation of the top five high-risk cancers. (**a**) Correlation between lung cancer and the other four cancers, (**b**) correlation between breast cancer and the other four cancers, (**c**) correlation between gastric cancer and the other four cancers, (**d**) correlation between colorectal cancer and the other four cancers, and (**e**) correlation between liver cancer and the other four cancers.

**Figure 7 entropy-23-00338-f007:**
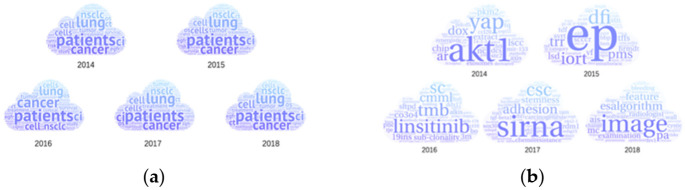
Hotspots of lung cancer were obtained by keyword trend analysis model (**a**) and improved keyword trend analysis model (**b**).

**Figure 8 entropy-23-00338-f008:**
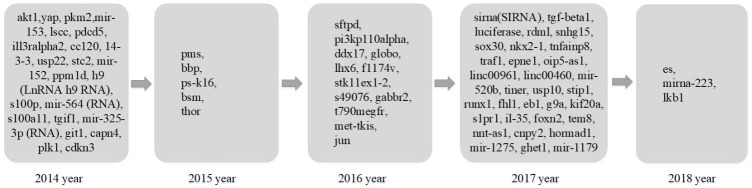
Research trends of lung cancer research related gene protein and invertase factor in the last five years.

**Figure 9 entropy-23-00338-f009:**
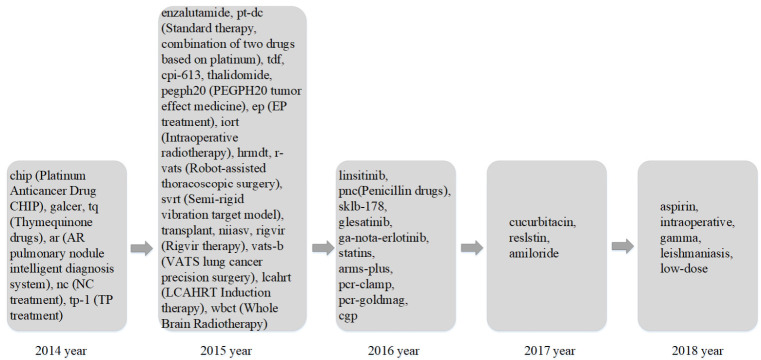
Research trends of lung cancer research related therapeutic drugs and methods in the last five years.

**Figure 10 entropy-23-00338-f010:**

Research trends of other related hotspots in the last five years.

**Table 1 entropy-23-00338-t001:** Number of research papers for the five cancers.

Cancer	2014	2015	2016	2017	2018
Lung	9322	9966	9446	9508	10,149
Breast	12,328	12,825	12,600	12,286	12,743
Gastric	3747	3572	3637	3414	3561
Colorectal	8950	9174	8778	8617	8868
Liver	6651	6871	6517	6431	6555

## Data Availability

Publicly available datasets were analyzed in this study. This data can be found here: https://www.ncbi.nlm.nih.gov/pmc/ (accessed on 20 November 2020).

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
