# Peer review of "Cancer Research Trend Analysis Based on Fusion Feature Representation"

_entropy, 2021, doi:10.3390/e23030338_

Round 1

Reviewer 1 Report

Cancer research has caught a lot of attention all over the world. And plenty of cancer research articles have been published. Therefore, automatically extracting knowledge from tens of thousands of articles to help the clinical doctors is of great significance. Natural language processing can provide useful method to Semantic feature learning is an important task in the field of natural language processing, and it is the basis of many downstream applications, such as text clustering and classification. In this paper, the authors propose two fusion algorithms and apply them to discover the research trends and the evolution of the hotspots for cancers. This idea is interesting, and the model has achieved good results. However, I think there are still some problems to be addressed.

First of all, among the machine learning models, why are these three methods chosen for combined?

Second, Figure 1 should be improved for it is not clear enough. Please double check other figures. The texts and words in the figures should be larger and clearer. We would like to see them more clearly.

Third, there are more than 100 cancers, in this paper, the authors explore five kinds of them. It is believed that the results are part of the facts.

In addition, Figure 5 and 7 can be described with numbers instead of graphs to show the results. Therefore, we can know more details.

In summary, I think that although this paper has a good idea, the design of the model is reasonable, the experiment is enough, and the content of the article is detailed enough.

Author Response

Response to Reviewer 1 Comments

Cancer research has caught a lot of attention all over the world. And plenty of cancer research articles have been published. Therefore, automatically extracting knowledge from tens of thousands of articles to help the clinical doctors is of great significance. Natural language processing can provide useful method to Semantic feature learning is an important task in the field of natural language processing, and it is the basis of many downstream applications, such as text clustering and classification. In this paper, the authors propose two fusion algorithms and apply them to discover the research trends and the evolution of the hotspots for cancers. This idea is interesting, and the model has achieved good results. However, I think there are still some problems to be addressed.

Point 1: First of all, among the machine learning models, why are these three methods chosen for combined? 

 Response 1: Cancer research has attracted much attention in the medical field. We aim to list the trend analyses of cancer research and find research hotspots. For this, using appropriate representation methods is an important task. The TF-IDF text representation model is an algorithm based on word frequency, which pays more attention to the number of times the words appeared in the document and does not consider the relative position between words. The TextRank model focuses more on the degree of co-occurrence between words in a fixed-length window. The TF-IDF and TextRank are classical representation methods and applied in many fields. Therefore, we choose these three methods for a multi-view of representation. We adopt the word2vec algorithm as the basic word representation method, capturing semantic and structural information. Further, the text is represented with the weighted product sums of words' representation. The weights are obtained by TF-IDF or TextRank.

Point 2: Second, Figure 1 should be improved for it is not clear enough. Please double check other figures. The texts and words in the figures should be larger and clearer. We would like to see them more clearly.

Response 2: Thanks for your suggestion. Figure 1 has been revised. We also checked other figures.

Point 3: Third, there are more than 100 cancers, in this paper, the authors explore five kinds of them. It is believed that the results are part of the facts.

Response 3: We selected five cancers as examples to list the research trend and hotspots. The chosen five cancers are the top 5 highest incidences in China in recent five years. The results of other cancers can be obtained with the proposed framework.

Point 4: In addition, Figure 5 and 7 can be described with numbers instead of graphs to show the results. Therefore, we can know more details.

Response 4: Thanks for your suggestion. We deleted Figures 5 and 7 and added description with numbers to lines 226 and 244, page 7.

Reviewer 2 Report

In this paper, two fusion algorithms are proposed, namely, the Tr‐W2v and Ti‐W2v algorithms. They are based the classical text feature representation model and consider the importance of words

What are the benefit of feature fusion? I would recommend to report accuracy, specify, and f-score

In addition, I would recommend to updated the related work with multi-modal fusion technique such as: (they used multimodal features from text, audio and visual and they used fusion technique)

Gogate, M., Dashtipour, K., Adeel, A. and Hussain, A., 2020. CochleaNet: A robust language-independent audio-visual model for real-time speech enhancement. Information Fusion63, pp.273-285.

Gogate, M., Dashtipour, K., Bell, P. and Hussain, A., 2020, July. Deep Neural Network Driven Binaural Audio Visual Speech Separation. In 2020 International Joint Conference on Neural Networks (IJCNN) (pp. 1-7). IEEE.

There are two types of fusion technique, early fusion and late fusion

Early fusion concatenate the features together and late fusion get prediction of each technique and combine them together and finally apply machine learning or deep learning. I would recommend to use these techniques as well.

Author Response

Response to Reviewer 2 Comments

In this paper, two fusion algorithms are proposed, namely, the Tr‐W2v and Ti‐W2v algorithms. They are based the classical text feature representation model and consider the importance of words.

Point 1: What are the benefit of feature fusion? I would recommend to report accuracy, specify, and f-score. 

 Response 1: The text representation in the classical method generally adopts the average of all word vectors. All words are considered equally important. This is obviously far-fetched because the importance of words in the text is different. The representation algorithms of TF-IDF and TextRank represent a text by calculating the weight of words in the text. The Word2Vec algorithm can capture the semantic information of words and does not consider the importance of words. The feature fusion can provide a multi-view of word representation. The silhouette coefficient is chosen as an evaluation index of clustering results. The accuracy and F-score are not considered for the unlabeled data.

Point 2: In addition, I would recommend to updated the related work with multi-modal fusion technique such as: (they used multimodal features from text, audio and visual and they used fusion technique)

Gogate, M., Dashtipour, K., Adeel, A. and Hussain, A., 2020. CochleaNet: A robust language-independent audio-visual model for real-time speech enhancement. Information Fusion, 63, pp.273-285.

Gogate, M., Dashtipour, K., Bell, P. and Hussain, A., 2020, July. Deep Neural Network Driven Binaural Audio Visual Speech Separation. In 2020 International Joint Conference on Neural Networks (IJCNN) (pp. 1-7). IEEE.

There are two types of fusion technique, early fusion and late fusion

Early fusion concatenates the features together and late fusion get prediction of each technique and combine them together and finally apply machine learning or deep learning. I would recommend to use these techniques as well.

Response 2: Thanks for your suggestion. We have updated the related work with multi-modal fusion technique in the Introduction section (See page 2, lines 57-62).

Reviewer 3 Report

In this manuscript, the authors proposed two text representation methods that improved Word2Vec method by combining it with TF-IDF or TextRank method. The results exhibited the proposed method was better than other traditional representation methods. 

The manuscript was well-organized and the attempt to improve Word2Vec was interesting. However, I have some significant concerns about the novelty of this manuscript comparing to previous works.

  1. The proposed idea that combines Word2Vec and TF-IDF or TextRank is not novel. I have found some papers very easily that propose very similar ideas. The authors must compare their method to the previous works and demonstrate what is different and improved.
    [Word2vec and TF-IDF]
    - Chen, Zhen. "Short text classification based on word2vec and improved TDFIDF merge weighting." 2019 3rd International Conference on Electronic Information Technology and Computer Engineering (EITCE). IEEE, 2019.
    - Zhu, Wei, et al. "A study of damp-heat syndrome classification using Word2vec and TF-IDF." 2016 IEEE International Conference on Bioinformatics and Biomedicine (BIBM). IEEE, 2016.
    [Word2vec and TextRank]
    - Zuo, Xiaolei, Silan Zhang, and Jingbo Xia. "The enhancement of TextRank algorithm by using word2vec and its application on topic extraction." Journal of Physics: conference series. Vol. 887. No. 1. IOP Publishing, 2017.

  2. Word2Vec is concrete and used broadly, but outdated method. If the main contribution of this manuscript was a research trend analysis, I think the authors can use advanced text representation methods such as BERT (Bidirectional Encoder Representations from Transformers),  or BioBERT that was trained with whole PubMed abstract and PMC full text data. The state-of-the-art language model represents the given text much better and will yield the improved cancer research trend analysis results.
    Devlin, Jacob, et al. "Bert: Pre-training of deep bidirectional transformers for language understanding." arXiv preprint arXiv:1810.04805 (2018).
    Lee, Jinhyuk, et al. "BioBERT: a pre-trained biomedical language representation model for biomedical text mining." Bioinformatics 36.4 (2020): 1234-1240.

Author Response

Response to Reviewer 3 Comments

In this manuscript, the authors proposed two text representation methods that improved Word2Vec method by combining it with TF-IDF or TextRank method. The results exhibited the proposed method was better than other traditional representation methods.

The manuscript was well-organized and the attempt to improve Word2Vec was interesting. However, I have some significant concerns about the novelty of this manuscript comparing to previous works.

Point 1: The proposed idea that combines Word2Vec and TF-IDF or TextRank is not novel. I have found some papers very easily that propose very similar ideas. The authors must compare their method to the previous works and demonstrate what is different and improved.
[Word2vec and TF-IDF]

- Chen, Zhen. "Short text classification based on word2vec and improved TDFIDF merge weighting." 2019 3rd International Conference on Electronic Information Technology and Computer Engineering (EITCE). IEEE, 2019.
- Zhu, Wei, et al. "A study of damp-heat syndrome classification using Word2vec and TF-IDF." 2016 IEEE International Conference on Bioinformatics and Biomedicine (BIBM). IEEE, 2016.
[Word2vec and TextRank]

- Zuo, Xiaolei, Silan Zhang, and Jingbo Xia. "The enhancement of TextRank algorithm by using word2vec and its application on topic extraction." Journal of Physics: conference series. Vol. 887. No. 1. IOP Publishing, 2017.

Response 1: Thanks for your comments. After carefully reading these three papers, we believe that there are some differences between our manuscript with the three papers. We found that most of the papers are aimed to build a classification model. From the view of machine learning, classification and clustering are two totally different models. The former needs classification labels to train a model, while the latter neither needs any label nor training. Clustering models can reveal the latent distribution of the data, and this is just what we want to do. Our purpose is to discover the trend of cancer research without human guidance. Besides, in our paper, we tried to improve the overall expression ability of text but not the angle of extracting keywords, which is different from others and is also our contribution.

Point 2: Word2Vec is concrete and used broadly, but outdated method. If the main contribution of this manuscript was a research trend analysis, I think the authors can use advanced text representation methods such as BERT (Bidirectional Encoder Representations from Transformers), or BioBERT that was trained with whole PubMed abstract and PMC full text data. The state-of-the-art language model represents the given text much better and will yield the improved cancer research trend analysis results.
Devlin, Jacob, et al. "Bert: Pre-training of deep bidirectional transformers for language understanding." arXiv preprint arXiv:1810.04805 (2018).
Lee, Jinhyuk, et al. "BioBERT: a pre-trained biomedical language representation model for biomedical text mining." Bioinformatics 36.4 (2020): 1234-1240.

Response 2: Thanks for your suggestion. Both Bert and BioBert are good methods. In the future work, we will introduce these models for a better representation.

Reviewer 4 Report

The authors proposed two fusion algorithms to learn the representation of the keywords. The authors showed the proposed fusion representations outperformed the classification models. Furthermore, trend analyses of cancer research were conducted based on the Tr‐W2v model. The writing is clear and easy to understand with the help of abundant tables and figures. While the contribution described in the manuscript is worthwhile, there are some limitations needed to be improved: 1) why the authors did not use TF-IDF + TextRank as the fusion model, 2) Replace Pseudocode in tables 1 and 2 with the equation, and put Pseudo code to supplementary, 3) Why use silhouette coefficient (SC) as SC is for the data without the label. Besides, SC should be introduced and cited, 4) the section of limitation is missing, 5) related work is missing, the works [1-4] are suggested to be cited for the different linguistic models.

[1] Hatzivassiloglou, V., Gravano, L. and Maganti, A., 2000, July. An investigation of linguistic features and clustering algorithms for topical document clustering. In Proceedings of the 23rd annual international ACM SIGIR conference on Research and development in information retrieval (pp. 224-231).

[2] Nam, S., Kim, S.K., Kim, H.G., Ngo, V. and Zong, N., 2016. Structuralizing biomedical abstracts with discriminative linguistic features. Computers in biology and medicine79, pp.276-285.

[3] Sarkar, K., 2009. Sentence clustering-based summarization of multiple text documents. TECHNIA–International Journal of Computing Science and Communication Technologies2(1), pp.325-335.

[4] Tang, B., Cao, H., Wang, X., Chen, Q. and Xu, H., 2014. Evaluating word representation features in biomedical named entity recognition tasks. BioMed research international2014.

Minor, Figures 1 and 5 are not readable. Figure 5 should re-scale in the y-axis to make it more readable.

Author Response

Response to Reviewer 4 Comments

The authors proposed two fusion algorithms to learn the representation of the keywords. The authors showed the proposed fusion representations outperformed the classification models. Furthermore, trend analyses of cancer research were conducted based on the Tr‐W2v model. The writing is clear and easy to understand with the help of abundant tables and figures. While the contribution described in the manuscript is worthwhile, there are some limitations needed to be improved:

Point 1: why the authors did not use TF-IDF + TextRank as the fusion model? 

Response 1: The representation algorithms of TF-IDF and TextRank represent a text by calculating the weight of words in the text. The Word2Vec algorithm can capture the semantic information of words and does not consider the importance of words. The feature fusion can provide a multi-view of word representation when combining word2vec with TF-IDF and TextRank.

Point 2: Replace Pseudocode in tables 1 and 2 with the equation, and put Pseudo code to supplementary.

Response 2: Thanks for your suggestion. We replaced the Pseudocode in tables 1 and 2 with related equations (see lines 141 and 155, page 4).

Point 3: Why use silhouette coefficient (SC) as SC is for the data without the label. Besides, SC should be introduced and cited.

Response 3: The silhouette coefficient (SC) is used for evaluating the clustering results. A reference has been added to introduce the SC (see line 226, page 7).

Point 4: the section of limitation is missing.

Response 4: Thanks for your suggestion. We have added a limitation discussion in the Conclusion Section (see lines 384-388, page 12).

Point 5: related work is missing, the works [1-4] are suggested to be cited for the different linguistic models.

[1] Hatzivassiloglou, V., Gravano, L. and Maganti, A., 2000, July. An investigation of linguistic features and clustering algorithms for topical document clustering. In Proceedings of the 23rd annual international ACM SIGIR conference on Research and development in information retrieval (pp. 224-231).

[2] Nam, S., Kim, S.K., Kim, H.G., Ngo, V. and Zong, N., 2016. Structuralizing biomedical abstracts with discriminative linguistic features. Computers in biology and medicine, 79, pp.276-285.

[3] Sarkar, K., 2009. Sentence clustering-based summarization of multiple text documents. TECHNIA–International Journal of Computing Science and Communication Technologies, 2(1), pp.325-335.

[4] Tang, B., Cao, H., Wang, X., Chen, Q. and Xu, H., 2014. Evaluating word representation features in biomedical named entity recognition tasks. BioMed research international, 2014.

Response 5: Thanks for your suggestion. The discussion of related work is added in the Introduction Section. The four related references have been cited (See page 2, lines 59).

Point 6: Minor, Figures 1 and 5 are not readable. Figure 5 should re-scale in the y-axis to make it more readable.

Response 6: Thanks for your suggestion. Figure 1 has been revised. Figure 5 is deleted and replaced with a description with numbers in lines 226-228, page 7. 

Round 2

Reviewer 3 Report

I understand the authors response that the previous works were different with respect to the task (e.g classification and clustering). However, I still have a concern that the proposed Tr-W2v and Ti-W2v algorithms should be compared to the recent language models. I do not know how much time was given for this revision (I feel it was very quick), but if possible I strongly recommend for the authors to apply the cutting-edge techniques to their algorithm. Minor comment: the citations of Bert and BioBert were missing at the limitation section.

Author Response

Response to Reviewer 3 Comments

Point 1: I understand the authors response that the previous works were different with respect to the task (e.g classification and clustering). However, I still have a concern that the proposed Tr-W2v and Ti-W2v algorithms should be compared to the recent language models. I do not know how much time was given for this revision (I feel it was very quick), but if possible I strongly recommend for the authors to apply the cutting-edge techniques to their algorithm. Minor comment: the citations of Bert and BioBert were missing at the limitation section.

Response 1: Thanks for your suggestion. We have added a Limitation Section (see lines 371-376, page 10). The citations of Bert and Biobert were added in the Limitation Section. The Bert and BioBert are suitable representation methods. We need to upload the revised file within 5 days, and the period is not enough to add the cutting-edge techniques to our methods. In future work, we will introduce these models for a better representation.

Reviewer 4 Report

The authors made the corresponding modifications based on the suggestions from the reviewers. The manuscript is improved while there remain some limitations: 1) Figure 6 is still not readable, 2)  limitations should be input prior to the conclusion and be more specific and comprehensive. Normally, limitations should cover all aspects, e.g., data, design of experiment, algorithm, etc. Authors should consider being more specific and concrete to the limitation. 3) reduce the conclusion to just summarize the study. Anything unique for contribution should move to the Introduction.

Author Response

Response to Reviewer 4 Comments

The authors made the corresponding modifications based on the suggestions from the reviewers. The manuscript is improved while there remain some limitations:

 Point 1: Figure 6 is still not readable.

Response 1: Thanks for your suggestion. Figure 6 has been updated.

Point 2: Limitations should be input prior to the conclusion and be more specific and comprehensive. Normally, limitations should cover all aspects, e.g., data, design of experiment, algorithm, etc. Authors should consider being more specific and concrete to the limitation.

Response 2: Thanks for your suggestion. We have added a Limitation Section (see lines 379-385, page 10).

Point 3: Reduce the conclusion to just summarize the study. Anything unique for contribution should move to the Introduction.

Response 3: Thanks for your suggestion. We revised the Conclusion Section and Introduction Section. The contribution is described in Introduction Section (see lines 65-73, page 2). The Conclusion Section is reduced to summarized the study (see lines 378-385, page 11).